# Essential Role of Granulosa Cell Glucose and Lipid Metabolism on Oocytes and the Potential Metabolic Imbalance in Polycystic Ovary Syndrome

**DOI:** 10.3390/ijms242216247

**Published:** 2023-11-13

**Authors:** Chen-Hua Zhang, Xiang-Yi Liu, Jing Wang

**Affiliations:** 1Queen Mary School, Medical College, Nanchang University, Nanchang 330006, China; jp4217120090@qmul.ac.uk (C.-H.Z.); jp4217120089@qmul.ac.uk (X.-Y.L.); 2Department of Cell Biology, School of Medicine, Nanchang University, Nanchang 330006, China

**Keywords:** glucose metabolism, lipid metabolism, granulosa cells, oocytes, PCOS

## Abstract

Granulosa cells are crucial for the establishment and maintenance of bidirectional communication among oocytes. Various intercellular material exchange modes, including paracrine and gap junction, are used between them to achieve the efficient delivery of granulosa cell structural components, energy substrates, and signaling molecules to oocytes. Glucose metabolism and lipid metabolism are two basic energy metabolism pathways in granulosa cells; these are involved in the normal development of oocytes. Pyruvate, produced by granulosa cell glycolysis, is an important energy substrate for oocyte development. Granulosa cells regulate changes in intrafollicular hormone levels through the processing of steroid hormones to control the development process of oocytes. This article reviews the material exchange between oocytes and granulosa cells and expounds the significance of granulosa cells in the development of oocytes through both glucose metabolism and lipid metabolism. In addition, we discuss the effects of glucose and lipid metabolism on oocytes under pathological conditions and explore its relationship to polycystic ovary syndrome (PCOS). A series of changes were found in the endogenous molecules and ncRNAs that are related to glucose and lipid metabolism in granulosa cells under PCOS conditions. These findings provide a new therapeutic target for patients with PCOS; additionally, there is potential for improving the fertility of patients with PCOS and the clinical outcomes of assisted reproduction.

## 1. Introduction

Human follicular development is a highly regulated process that depends on the mutual coordination of oocytes and all the cell types around them. Among them, granulosa cells proliferate and enclose oocytes in the follicle before ovulation, playing an important role in the normal development of oocytes before ovulation and the subsequent fertilization process. Significant glucose metabolism and lipid metabolism can occur in granulosa cells, and a large number of energy substrates and hormones are produced during these processes. The efficient substance exchange network between oocyte and granulosa cells—which is constructed with a paracrine, a gap junction, a contact junction, and extracellular vesicle transport—can transport nutrients and exchange signals and can effectively compensate for the energy gap that occurs during oocyte development. This process also realizes the coordination of oocyte development and the follicle maturation cycle through changing the levels of signaling molecules and energy supplies. Therefore, disturbances of glucose and lipid metabolisms in granulosa cells are bound to lead to impaired reproductive function.

Polycystic ovary syndrome (PCOS) is an important cause affecting female fertility, with an incidence of 6–21% among women worldwide [1]. The main manifestations are ovarian enlargement and a large number of cysts, accompanied by irregular menstruation, elevated androgen levels, and ovulation disorders. A variety of experiments have shown that abnormal glucose metabolism and lipid metabolism in granulosa cells are related to PCOS; these affect insulin resistance, mitochondrial function, steroid hormone synthesis and processing, and other important links affecting follicular development. This article focuses on the unique material exchange and information communication between granulosa cells and oocytes, the important role of glucose and lipid metabolisms in granulosa cells in the development of oocytes, and the relationship between pathological glucose and lipid metabolism and PCOS. This article can help provide new targets for PCOS drug development and provide new information for clinicians to refer to when diagnosing and treating PCOS.

## 2. Follicular Growth and Oocyte Maturation

### 2.1. Follicular Growth

The ovary is the main female reproductive organ, and its function mainly depends on follicle cells located in the ovarian cortex. Follicle cells show structural and functional diversity at different stages of development. Folliculogenesis first occurs in the fetus at the age of four months, where primordial follicles begin to build and accumulate to generate the resting follicle pool (RF) [2]. Primordial follicles have a simple structure, containing only oocytes that stay in meiosis I and a single layer of squamous granulosa cells (GCs) covering their surface; they are confined to the basement membrane [3]. Subsequently, primordial follicles undergo environmental selection, leading to their transformation into primary follicles, with major changes including the cubization of peripheral squamous granulosa cells [4] and the appearance of a glycoprotein dissection (zona pellucidum) between primary oocytes and granulosa cells—which, at this stage, is mainly used to support GC and oocyte communication [5].

Primary follicular development constantly stacks peripheral granulosa cell layers; through the connective tissue recruiting theca cells covering the basement membrane, the theca cells undergo a divide into two layers, the theca interna and the theca externa. The cells in theca interna have stem cell function and can achieve membrane regeneration, and the cells in theca externa enact differentiation, mainly on good barrier function [2,6]. At the same time, primary follicles form cavities between granulosa cells during development, called antral follicles. Prior to puberty, the ovary lacks hypothalamic–pituitary–ovarian axis stimulation, and the primary follicle is maintained for a period of time before dying naturally. After puberty, under the action of follicle-stimulating hormone (FSH) secreted by the hypothalamic–pituitary–ovarian axis, some primary follicles escape death and transform into secondary follicles, also known as dominant follicles. The main manifestations include the union and expansion of the fluid space between granulosa cells into sinusoids, and the differentiation of granulosa cells into two subsets: cumulus cells (CCs), located around the oocyte; and mural granulosa cells (MCGs), located outside the sinus [7]. Subsequently, CCs closely attached to the oocyte further differentiate to form corona radiata, which can support and guide sperm into the egg during fertilization.

In the final stage of follicular development, the positive feedback of the hypothalamic–pituitary–ovarian axis mediates the surge of luteinizing hormone (LH) secretion, leading to the release of oocytes from the follicle and the activation of their meiosis, and the arrest in the middle of meiosis II [8]. In addition, during the preovulatory period, the cumulus oocyte complex further processes hyaluronic acid (HA), synthesized by MCGs and CCs, to construct a highly viscous extracellular matrix; this matrix can promote the expansion and exit of the cumulus oocyte complex [9,10] and mediate sperm penetration into the oocyte during fertilization. The expelled oocyte complex moves through the inner wall of the ovary and attaches to the fallopian tube to wait for fertilization. The peripheral cumulus cells can secrete chemoattractants such as progesterone to efficiently attract sperm [11]. Ruptured follicles that have lost their oocytes are called corpus luteum. Residual GCs are transformed into large luteal cells and produce progesterone, and residual follicular membrane cells are transformed into small luteal cells and secrete estrogens and androgens. The precise regulation of multiple hormones enables the maintenance of pregnancy under fertilization conditions, as well as the degeneration of the corpus luteum and the activation of the next menstrual cycle under non-fertilization conditions [11,12].

### 2.2. The Role of Granulosa Cells in Mature Oocytes

Despite the fact thar they are both derived from the same progenitor cell precursors, MGCs and CCs have significant functional differences [13]. MGCs are distant from the cumulus–oocytes complex (COC) and spaced apart from the fluid-rich antral layer. They secrete hormones and achieve oocyte regulation through antral fluid transport [7]. During the early stages of oocyte development, the MCGs are mainly responsible for the synthesis of estrogen (estradiol), which is used for the development of female secondary sex characteristics and the regulation of the reproductive cycle [14]. Prior to ovulation, granulosa cells undergo a surge in progesterone secretion, which regulates cellular communication and metabolic cooperation between oocytes and CCs and activates the expression of COC and oocyte maturation-related genes to accelerate ovulation [15,16]. It can also lead to follicular maturation-related morphological changes such as stigma formation [17]. CCs are primarily responsible for providing environmental support for oocyte maturation. They establish gap junctions with the oocyte by forming long cellular projections that can cross the zona pellucida through cellular metaplasia [18]. These gap junctions allow oocytes through and permit the two-way conduction of information between granulosa cells, enable metabolism and nutrient exchange, and then control the development of the COC [19,20]. Due to the dynamic nature of the development of oocytes, CCs play different functions at different stages of oocyte development. The nuclei and cytoplasms of early-stage oocytes develop synchronously. Decompaction of the chromatin structure in oocytes allows for substantial RNA and protein synthesis [21]. At the same time, the inner membrane system undergoes complex reorganization and forms a vesicle-like structure surrounding the nucleus—this formation is called the germinal vesicle stage (GV)—which reserves vitamins, proteins, nucleic acids, and other metabolic raw materials for oocyte metabolism [22]. CCs are mainly responsible for oocyte protection at this stage. Prior to oocyte maturation, chromatin in the GV condenses and surrounds the nucleoli in preparation for meiosis, thus decreasing the transcriptional and metabolic activity of the oocyte. CCs provide nutritional support for this stage, as well as for signal transduction. Among them, nucleotides, lipids, glucose (pyruvate), cholesterol, etc., provided by CCs cells, contribute to germinal vesicle breakdown (GVBD) and promote the progression of meiosis [23,24,25].

## 3. “Bidirectional Regulation” of Granulosa Cells and Oocytes

In the process of follicular development, the oocytes are the regulatory object of the granulosa cells; importantly, however, the granulosa cells can be regulated by the oocytes. This “bidirectional interaction” is mainly achieved through four pathways, including the gap junction mentioned above, the receptor- and ligand-dependent paracrine pathway, and the newly discovered adhesion junctions and extracellular vesicle (EV)-mediated macromolecular transport.

### 3.1. Paracrine Pathway

A majority of several currently elucidated paracrine signaling molecules are secreted by oocytes, as shown in Figure 1. These include bone morphogenetic proteins (BMPs) and growth differentiation factors (GDF9), which bind to transforming growth factor beta (TGFβ) family receptors and activate downstream phosphorylation cascades [26,27,28]. Among them, the *BMP15* gene is located on the X chromosome and encodes a secreted protein that binds to and dimerizes two TGFβ receptor subunits, ALK6 and BMPRII [29,30]. The dimerization receptor presents an activated state and phosphorylates downstream proteins of the Sma- and Mad-related protein (SMAD) family; then, the SMAD complex translocates to the nucleus and interacts with specific transcription factors to achieve specific transcription [30,31,32]. GDF9 has a similar mechanism of action as that of BMP15, but it is encoded by an autosome and activates different dimeric receptors (ALK5 and BMPRII) to achieve different SMAD activation and transcriptional regulation [33,34,35]. In addition, a heterodimer named cumulin was recently identified; it is composed of GDF9 and BMP15 proteins and activates the downstream BMPRII-ALK4/5/7-ALK6 complex [30,36]. This phenomenon of ligand dimerization suggests that paracrine factors have synergistic features in function.

The functions of BMP15 and GDF9 are mainly focused on five aspects: oocyte development regulation, intercellular communication regulation, hormone secretion regulation, resistance to granulosa cell apoptosis, and protein expression regulation. For oocyte development, BMP15 and GDF9 activate the hyaluronic acid synthesis pathway by upregulating *Has2* and *PTX3* genes to achieve the secretion and accumulation of the extracellular matrix and promote cumulus expansion and ovulation [30,37]. A mouse-based study showed that GDF9 can also participate in the differentiation of GCs into CC in preantral follicles [7]. For intercellular communication, BMP15 and its family members BMP4 and BMP7 inhibit gap junctions through the BMP4/7/15-Cx43 pathway. This inhibition may reduce the transmission of meiotic inhibitors through gap junctions in oocytes, promote meiosis, and thus induce GVBD [38,39]. One study knocked down GDF9 by RNAi interference and found that transzonal projections (TZPs), the basic structures used to build gap junctions, were significantly reduced. This suggests that paracrine molecules such as GDF9 may act as chemokines for TZP growth [40]. For hormone secretion, BMP15 and its family members are dependent on acute phase regulatory protein (StAR) to reduce progesterone production, thereby inhibiting luteinization. Meanwhile, GDF9 could further inhibit progesterone secretion by inhibiting pituitary gonadotropin. This phenomenon can be reversed by low *BMP15* expression in the luteal phase [37]. In addition, GDF9 has been shown to stimulate androgen synthesis and thus primordial and primary follicle development in rats [41]. For the resistance to granulosa cell apoptosis, BMP15 and BMP6 are secreted by oocytes and establish a molecular gradient. Cumulus cells are in a high-BMP environment for a long time, which leads to a decrease in the expression of pro-apoptotic genes (*BAX*, *CASP9*, and *TP53*) and an increase in the expression of anti-apoptotic genes (*BCL-2*) [42]. For the regulation of protein expression, BMP15 and GDF9 are mainly involved in the regulation of KIT ligand (KITL), a paracrine factor secreted by granulosa cells. KITL contains two isoforms, KITL1 and KITL2 [43]. They regulate oocyte survival, activation, and apoptosis by activating the downstream P13K-PIP3-AKT-FKHRL1 pathway; this is achieved through the activation of tyrosine kinase receptors (KITs), located on oocyte membranes [44]. Before puberty, when follicles are not stimulated by FSH, KITL2 plays an important role in primordial follicle formation and development. FSH stimulates granulosa cells after puberty, leading to an increase in the ratio of KITL1/2, which restricts the development of oocytes and tends to gradually mature [45]. Studies have shown that a negative feedback loop can be formed between BMP15 GDF9 and KITL, and BMP15 can promote the expression of KITL1 and KITL2 and maintain the relative stability of KLTL1/2, which helps to improve the sensitivity of GCs to FSH and promote the growth and development of primordial oocytes [45]. At the same time, high concentrations of KITL can inhibit the expression of *BMP15* [45]. For GFP9, GFP9 inhibits *KITL* expression, but this inhibition is only effective for fully developed oocytes, and it has no significant effect on developing oocytes [46]. In conclusion, GDF9 and BMP15 have similar structures and paracrine characteristics, but their functions either serve to promote each other or are antagonistic to each other. At the same time, it can rely on negative feedback to achieve strict regulation of its own expression.

### 3.2. Gap Junction

Gap junctions mainly include homologous gap junctions between granulosa cells and heterologous gap junctions between granulosa cells and oocytes. Heterologous gap junctions are particularly important for oocyte functional support. This special intercellular communication depends on special structural changes of granulosa cells called transzonal projections (TZPS) [47], as shown in Figure 2. The structure of TZP is similar to that of filopodia, and it relies on the actin skeleton to form long cellular processes that cross the zona pellucidum and bulge at the end to increase the contact area between the oocyte and TZP [48,49]. Gap junctions rely on the formation of connexin hexamers on the surface of the two cells, which link to each other and form a central channel for small molecules to pass through, realizing the direct exchange of substances between cells. E-cadherin in oocytes and N-cadherin on granulosa cells contact each other to form stable adhesive junctions to stabilize gap junctions [50,51].

TZP production is mainly dependent on granulosa cells, and there are two mainstream TZP generation models. In the early stage, TZP may be passively generated during the production of zona pellucidum. The mechanism is that the production of zona pellucidum leads to the separation of granulosa cell and oocytes, but the preexisting adhesion link sites of the two cells do not separate and stretch the cytoplasmic structure of granulosa cells, resulting in their elongation [20,52]. Subsequent TZP is synthesized de novo by actin [53]. The growth regulation of TZP in granulosa cells by oocytes is described above, but the molecular regulation of TZP by granulosa cells is yet to be fully elucidated. Myo10 has been shown to inhibit TZP function, but no cases of developmental defects caused by Myo10 inhibition have been found, so this factor may be nonessential for TZP development [41,54]. In addition, embryonic (A)-binding protein (EPAB) and proline-rich tyrosine kinase (PTK2) can affect the function of E-cadherin, leading to structural instability and a reduction in TZP [55]. The disappearance of TZP is mainly stimulated by external hormones. The preovulatory surge in luteinization (LH) results in the release of epidermal growth factor (EGF), which binds to the cumulus cell surface receptor EGFR. This activates a downstream signaling cascade that activates calpain to cleave the adhesion junctions between granulosa cells and oocytes. TZP can be recycled to granulosa cells by inducing actin skeleton reorganization [55,56,57].

The function of gap junctions is mainly divided into two parts: providing metabolic support for oocytes and regulating oocyte development and maturation. The lack of enzymes or transporters involved in protein, lipid, and carbohydrate metabolic pathways in oocytes has not been acquired through the long process of evolution of the body but is compensated by the provision of relevant enzyme metabolites by granulosa cells. Amino acids (glycine, alanine, and proline) are common amino acid metabolic pathway supplements due to the lack of membrane transporters, such as SLC7A6 and SLC38A3, involved in exogenous amino acid uptake in oocytes [58]. In terms of glucose metabolism and lipid metabolism, due to the lack of related enzymes in glycolysis and cholesterol synthesis pathways, oocytes are unable to produce pyruvate, cholesterol, and other products related to the glucose and lipid metabolism pathways [55]. Therefore, a substrate supply pathway synthesized by granulosa cells and transported by gap junctions is required for glucolipid metabolism to be possible in oocytes. The relationship between granulosa cell glucose and lipid metabolism and oocytes is described in detail below. During oocyte development, the progression of meiosis is regulated by gap junction transport substances. After activation in meiosis II, oocytes need to be arrested in metaphase II of meiosis and await fertilization. Granulosa cells release a large amount of cAMP and enter oocytes through the gap junction. The high concentration of cAMP in oocytes downregulates the activity of cell cycle-dependent kinase (CDK), leading to meiotic arrest [57,59]. Through gap links, the granulosa cells also transport cGMP, which is an inhibitor of camp-specific phosphodiesterase (PDE) 3A that effectively inhibits cAMP hydrolysis in oocytes and enhances meiotic inhibition. Oocytes are exposed to high concentrations of LH prior to ovulation, and LH activates the downstream pathway of EGFR to cause a plunge in the intracellular concentration of cGMP, which activates the originally inhibited (PDE) 3A and hydrolyzes cAMP, resulting in the resumption of meiosis in oocytes [60]. Meanwhile, the reduction in the cAMP supply, caused by TZP retraction, further reduces cAMP concentration. In summary, oocytes are subject to efficient regulation by granulosa cell gap junctions for a long time, which is essential for the normal maintenance of ovarian function.

### 3.3. Adhesion Connection and Extracellular Vesicles (EVs)

Adhesion junctions located at the TZP, the only site of contact between granulosa cells and oocytes, play an essential role in the information exchange between the CCs and oocytes. With the development of research, it has been found that the information exchange between TZPS is not limited to gap junctions. Two interesting modes of intercellular communication, contact junction-mediated and extracellular vesicle (EV)-mediated macromolecular transport, have been found to play important roles in the communication between granulosa cells and oocytes, as shown in Figure 2.

Initially, it was found that contact junctions were involved in the “stretching” behavior of granulosa cells during the establishment of primitive TZPS, and the involvement of two common cell surface cadherins (E-cadherin and N-cadherin) was noted [20,52]. The subsequent discovery of several other adherens junctions has changed our understanding of them. Previously considered to be a paracrine factor of granulosa cells, KITL has been shown to bind to the cell membrane and combine with KIT to play a similar role as that of the paracrine factor [61]. This adhesion junction relies on its highly stable structure to play a more stable role in the regulation of oocytes than paracrine signals [62]. The Notch–Jagged junction is another type of adhesive junction; Notch is present on granulosa cells, and Jagged is present on oocytes. In one experiment, Jagged was knocked out by targeting, and it was found that Jagged knockout individuals were unable to undergo granulosa cell thickening during follicular development [63]. A control experiment among patients with polycystic ovary syndrome (PCOS) showed that, compared with individuals without PCOS, the eggs of patients with PCOS extracted high Notch-2 in oocyte complexes; here, *Notch-3*, *Jagged-1*, and *Jagged-2* gene expression decreased significantly. Although the specific function of this connection has not been elucidated, it is certain that it is necessary for the normal development of COC [64].

Gap junctions at TZP tips are only limited to small molecule transport, while the transfer of large molecules such as mRNA may depend on intercellular signaling mediated by extracellular vesicles. Among them, extracellular vesicles are mainly divided into two types according to their structure and synthesis pathways [65,66]. Microvesicles are relatively large in diameter (100–1000 nm) and are produced by budding from the cytoplasmic membrane. Exosomes are relatively small in diameter (50–200 nm) and originate from nuclear endosomes. After complex intracellular screening and processing, they gradually mature and are stored in the more multivesicular body (MVB); its release is dependent on the MVBand plasma membrane fusion. Several experiments have been used to support this hypothesis. When granulosa cells containing green fluorescent protein (*GFP*) expression were co-cultured with normal oocytes, green fluorescence could be detected in the oocytes, and GFP-related mRNA could be extracted, indicating that GFPmRNA from granulosa cells could be transferred to oocytes through a yet-unknown mechanism [49]. Real-time imaging technologies have been used to observe TZP mRNA in active two-way transportation [49,67]. This further supports the hypothesis that mRNA is transported by extracellular vesicles. In addition, current studies mainly focus on demonstrating the effect of vesicle transport produced by granulosa cells on oocytes. It remains to be elucidated whether this signaling is “bidirectional”, or whether oocytes can regulate granulosa cells through EVs. Several studies have confirmed that granulosa cells can absorb EVs and their function through the regulation of EVs. Researchers loaded fluorescent markers into EVs and co-cultured them with granulosa cells; they found fluorescent markers in granulosa cells, indicating that granulosa cells have the potential to fuse with EVs [68,69]. Meanwhile, studies have found that EVs in follicles can induce CCs’ expansion, which is based on the principle that specific mRNAs in EVs are introduced into granulosa cells, and the mRNAs express and synthesize transcription factors to regulate the expression of CCs expansion genes [70,71]. The bidirectional flow of mRNA in TZP can also indirectly prove that EVs can mediate mRNA uptake in granulosa cells [49,67]. However, no study has addressed the issue of whether oocytes are involved in the secretion of EVs in follicular fluid.

## 4. Glucose and Lipid Metabolism in Granulosa Cells and Oocytes in Healthy Women

### 4.1. Characteristics of Energy Metabolism in Granulosa Cells and Oocytes

The oocyte is the largest mammalian cell in the body, meaning that it has a relatively small surface area for material exchange. Glucose and lipid metabolism are the two most important energy sources in cells. Maintaining the stability of glucose and lipid metabolism in oocytes is of great significance for follicle and future embryo development. The “bidirectional regulation” of granulosa cells and oocytes compensates for this deficiency, and granulosa cells make efficient development possible by continuously “supplying” the oocyte. Taking glucose metabolism as an example, granulosa cells mainly rely on glycolysis to meet their energy requirements, while oocytes rely more on mitochondrial oxidative phosphorylation [24,72]. In the case of granulosa cells, excess glycolysis can generate the redundancy of pyruvate and lactate, which can flow into the mitochondria of the oocyte through gap links and serve as feedstock for oxidative phosphorylation. The dependence of oocytes on oxidative phosphorylation is mainly due to the lack of multiple enzymes in their upstream metabolic pathways, which deprive them of many of their own energy synthesis pathways. However, this is not a bad thing for the oocyte, and this deficiency promotes oocyte energy metabolism away from dependence on a single energy source. The transcellular transport of granulosa cells, uptake of metabolic materials in the surrounding medium by oocytes, and glucose and lipid metabolism of oocytes themselves can all provide raw materials for oxidative phosphorylation. This cooperation of energy metabolism is not static, and the different developmental patterns of oocytes at different stages imply diversification of energy requirements. The oocyte itself can change the efficiency of oxidative phosphorylation by changing the expression of electron transport chain-related protein (ETC) [73,74]. The regulation of energy by granulosa cells is more diverse: before oocyte maturation, TZP retraction of granulosa cells interrupts gap junctions, which can transiently reduce oocyte cAMP and regulate cell cycle [57]. The normobaric oxygen concentration of preovulatory follicles is usually very low, and hypoxia can activate glycolysis and oxidative phosphorylation by upregulating the *HIF1* gene, thus ensuring the energy supply of oocytes [75]. At the same time, granulosa cells are constantly regulated by paracrine factors (BMPs and FGF8) secreted by oocytes to meet the specific energy demand of oocytes. For example, on the eve of ovulation, oocytes promote the glycolysis of granulosa cells through paracrine factors [76]. Therefore, it is of great significance to understand the characteristics of glucose and lipid metabolism in granulosa cells and oocytes, especially the effect of granulosa cell metabolism on oocytes, for the study of ovarian development and the treatment of related diseases. Granulosa cell and oocyte metabolic pathways are shown in Figure 3.

### 4.2. Effect of Granulosa Cell Glucose Metabolism on Oocytes

#### 4.2.1. Glycolysis

Glucose is the most abundant energy substrate in the human body, and the cumulus–oocyte complex is also highly dependent on glucose metabolism. This metabolism is mainly provided by glycolysis and its downstream pathways. In contrast, the efficiency of glucose metabolism in oocytes is low because of the lack of glucose transporter (GLUT4) on the cell surface and the lack of phosphofructokinase (PFK) in the cells [77]. PFK is an important enzyme in glycolysis. The loss of PFK can lead to the failure of glucose to be converted to pyruvate and interrupt glycolysis. Granulosa cell mono-oocytes are distinct in that they possess high PFK activity and abundant GLUT transporters. Under normal conditions, FSH can activate glycolysis through the AMPK-SIRT1 pathway, which leads to efficient glucose uptake by granulosa cells; its glycolytic metabolites, pyruvate and lactate, are secreted in large amounts and rapidly delivered to oocytes through gap junctions. Oocytes make full use of glycolytic products to obtain energy through the tricarboxylic acid cycle and oxidative phosphorylation [78].

Granulosa cells have high glucose uptake and metabolic efficiency due to their high enzymatic activity [79]. However, in granulosa cells, most of the glucose is arrested after glycolysis; that is, pyruvate produced by glucose does not undergo subsequent tricarboxylic acid cycle (TCA) and oxidative phosphorylation, but it is excreted as pyruvate or lactate and supplied to the oocyte. This phenomenon is mediated by an enzyme called pyruvate dehydrogenase kinase (PDK) [80]. PDK is highly expressed in cumulus cells and inhibits pyruvate dehydrogenase (PDH), a key enzyme that controls pyruvate entry into the tricarboxylic acid cycle (TCA) [80]. The reason that granulosa cells have evolved this mechanism may be to reduce the damage caused by oxidative stress during metabolism. Because there are a large number of immature mitochondria in the process of follicular development, the tricarboxylic acid cycle (TCA) and oxidative phosphorylation can lead to the activation of a large number of immature mitochondria and produce a large amount of ROS [81]. ROS can damage DNA and a variety of cellular structures, thereby reducing the number of mitochondria, inhibiting their activity and function, and greatly reducing the efficiency of energy metabolism. Despite the dominant role of gap junctions in the transport of glycolytic material, interesting phenomena were found during oocyte development: TZP between granulosa cells was greatly reduced, and gap junctions were disrupted prior to oocyte maturation. Thus, the cell-to-cell transmission of cAMP concentration and other meiotic inhibitors is blocked, activating meiosis [38,39]. However, the energy metabolic requirements of oocytes in the mature stage are twice as high as before [76], and the chromatin condensation before the meiosis of oocytes leads to a significant reduction in their transcriptional and metabolic activities; therefore, the external energy support is needed more. In addition, before ovulation, granulosa cells heavily activate glycolysis in response to hypoxic stimulation and paracrine factors from oocytes [75]. Therefore, we speculate that energy metabolism between oocytes and granulosa cells is no longer critically dependent on gap junctions but is likely to be highly dependent on diffusion within the follicle just before ovulation. Several studies have supported this view in different ways. By culturing granulosa-cell–oocyte complexes in vitro, pyruvate and lactate concentrations in the culture medium were found to surge before ovulation. This is consistent with the idea that glycolytic products are largely propagated through gap junctions at this time [82]. At the same time, the pyruvate and lactate transporter known as monocarboxylic acid transporter (MCT) was found to be highly expressed on the oocyte surface, contributing to the oocyte uptake of energy precursors in the follicular fluid [82,83].

Lactate plays an important role in COC metabolism. Lactate usually serves as a marker of anaerobic metabolism in mammals, but in COC, lactate acts more as a “bridge” between aerobic glycolysis in granulosa cells and oxidative phosphorylation in oocytes. This bridge contains two enzymes with opposite functions: lactate dehydrogenase A(LDHA) in granulosa cells and lactate dehydrogenase B(LDHB) in oocytes [79]. LDHA in granulosa cells partially converts pyruvate produced by glycolysis to lactate and transfers it to the oocyte, and the LDHB present in the oocyte converts lactate again to pyruvate for energy supply. Although the formation of this pyruvate–lactate–pyruvate pathway has not been fully elucidated, the most important reason may be to adapt to the hypoxia state in the late follicular phase. The upregulation of *LDHA* expression induced by hypoxia-inducible factor (HIF) confirms this idea [84]. The maintenance of lactate metabolism by oocytes in the presence of high oxygen concentration is known as the Warburg effect, and this metabolic mechanism is commonly seen in cancer cells to meet their proliferation in the hypoxic environment [85,86]. In addition, this lactate-dependent mode of energy transport apparently increases the storage of carrier proteins on the cell surface. When gap junctions are not available, lactate and pyruvate receptors on the oocyte surface can be activated simultaneously to achieve the rapid uptake of energy substrates.

Pyruvate and lactate undergo completely different patterns of cellular energy metabolism after uptake by oocytes and in granulosa cells. Compared with the highly expressed PDK in cumulus cells, the low PDK level in oocytes activates a large amount of PDH, which catalyzes the conversion of pyruvate to acetyl-CoA, the starting substance of TCA [80]. TCA and oxidative phosphorylation produce about 18 times more ATP than glycolysis does in one cycle, which provides sufficient energy support for oocyte development and normal metabolism [87]. The current regulation of granulosa cell glycolysis by oocytes has not been demonstrated in humans. A mouse-based study has shown that oocyte regulation of granulosa cells occurs through the mechanisms of two paracrine factors, BMP15 and FGF8. Neither of them can achieve the glycolytic regulation of granulosa cells when they are alone [88]. Further studies have shown that this mechanism was achieved through the synergistic effect of FGF8 and BMP5 to upregulate the expression of *ALDOA*, *ENO1*, *LDHA*, and other glycolysis-related enzymes [88].

#### 4.2.2. Pentose Phosphate Pathway

In the process of mitochondrial oxidative phosphorylation in oocytes, ROS is inevitably produced as a by-product. ROS will have a series of side effects on follicular development, including accelerating oocyte senescence and limiting oocyte maturation and blastocyst formation [89,90]. Therefore, oocytes and cumulus cells have evolved anti-ROS mechanisms, which are mainly mediated by glutathione (GSH), provided by the pentose phosphate pathway (PPP) [91]. During glycolysis, a small fraction of glucose 6-phosphate is oxidized by glucose 6-phosphate dehydrogenase and enters the PPP pathway. Two metabolites, nicotinamide adenine dinucleotide phosphate (NADPH) and ribose 5-phosphate, are formed after undergoing oxidative and nonoxidative stages. Among them, NADPH can reduce glutathione disulfide (GSSH) to GSH to maintain intracellular GSH stores [91]. However, this antioxidant mechanism has certain limitations. Although there is a surge in GSH production in cumulus cells when oocytes are about to mature, its detoxification effect is very limited, and it can only neutralize a low concentration of reactive oxygen toxicants [92]. Therefore, the stabilization of the ambient oxygen concentration is still required to prevent oxidative stress in oocytes. Furthermore, based on the variability of PPP metabolism in granulosa cells, NAPPH in oocytes, like pyruvate and lactate, is highly dependent on transcellular transport in granulosa cells. Another function of the PPP is to support oocyte meiosis. 5-phosphoribosyl, a metabolite of the PPP pathway, can be used to synthesize phosphoribosyl pyrophosphate (PRPP), which is an important intermediary for the de novo and salvage synthesis of nucleotides [93]. NADPH can also be used as a raw material for the synthesis of nucleotides and fatty acids. Therefore, the PPP pathway can provide sufficient raw material reserve for oocyte meiosis. Studies have shown that inhibition of PPP enzyme activity in an in vitro culture model of porcine COC can significantly reduce the meiotic rate of oocytes [94].

Although the PPP pathway does not produce ATP, it was found to be indirectly involved in energy regulation by inhibiting glucose uptake and lactate production in the glycolytic pathway [95]. In addition, the PPP pathway is essential for the developmental competence of oocytes. Many mammal-based experiments have demonstrated that oocyte nuclear and cytoplasmic maturation depends on PPP [96,97]. Recent studies have shown that this process may be regulated by the 5-phosphoribose–NADB–Sirtuin pathway [98]. NAD+ is a ubiquitous enzymatic cofactor in cells, and Sirtuins represent the lysine deacetylation kinase family. High PPP levels generate large amounts of ribose 5-phosphate, and its metabolite PRPP acts as a precursor of NAD+ to induce massive NAD+ synthesis. High levels of NAD+ activate Sirtuin activity to coordinate organism metabolism and aging processes [98]. In a mouse-based experiment, researchers utilized the delivery of NAD+ precursors or genetic engineering to overexpress Sirtuin; this led to significantly restored fertility in aged mice [99].

#### 4.2.3. Hexosamine Biosynthetic Pathway

The hexosamine biosynthesis pathway (HBP) is another glucose metabolism pathway, and the final product is UDP-N-acetylglucose. Most of this UDP-N-acetylglucose is used to create the hyaluronic acid intercalation between granulosa cells and oocytes, promoting the synchronous growth of the extracellular matrix during cell development. A small fraction binds O-linked glycosyltransferases to glycosylated serine/threonine residues of specific proteins, thereby altering cellular fuel sensing and signaling [100,101]. Under normal circumstances, HBP accounts for only 5% of the total glucose consumption [102]. In the swelling stage of cumulus, oocytes need to rely on hyaluronic acid to obtain large amounts of nutrients, and rely on the expansion of hyaluronic acid to dispose of the shackles of granulosa cells. Therefore, the proportion of HBP glucose consumption in granulosa cells increased significantly at this stage, reaching 25% [103]. These metabolic changes mainly depend on the cooperation between the endocrine system and oocyte paracrine. During ovulation, the LH surge activates the LH-EGF pathway to stimulate cumulus expansion and the induction of oocytes, while oocyte paracrine factors enhance the sensitivity of cumulus cells to EGF [104]. Under the abovementioned stimulation, granulosa cells express large amounts of hyaluronic acid synthetase (HAS2), pentraxer-related protein (PTX3), and other essential raw materials for hyaluronic acid synthesis, and they activate the HBP pathway [86]. In addition, considering the important role of hyaluronic acid in fertilization and early embryonic development, the loss of HBP may have serious consequences for fertilization and fetal development. However, current studies have found that HBP is not a reliable indicator of fertilization success [105].

### 4.3. Effect of Granulosa Cell Lipid Metabolism on Oocytes

#### 4.3.1. Fatty Acid Metabolism

Lipid metabolism is another metabolic pathway in granulosa cells and oocytes, and the high variability in the number and size of lipid droplets in oocytes during oocyte growth under the microscope implies that it is closely related to oocyte function [106]. Lipids are mainly metabolized in the form of fatty acids. In oocytes, most fatty acids are stored as triacylglycerol (esterified form of fatty acids) in cytoplasmic lipid droplets, and a small part is free in the cytoplasm [107]. Lipid droplets are vesicular structures that are rich in perilipin (PLIN). Under normal conditions, PLIN and comparative gene identification 58 (CGI-58) bind to maintain the stability of the vesicle structure. External stimuli can cause the phosphorylation of PLIN and lead to vesicle instability and rupture. The released triglyceride binds to hormone-sensitive lipase (HSL), which is phosphorylated at the same time as PLIN and is then decomposed into fatty acids [108]. Meanwhile, CGI-58 and phosphorylated PLIN isolate and activate adipose triglyceride lipase (ATGL), further promoting lipid metabolism [108]. Fatty acids are directly metabolized to acetyl-CoA through A pathway known as β-oxidation, which converges with the glucose metabolism pathway to enter the tricarboxylic acid cycle and enable oxidative phosphorylation. Individual fatty acids can be metabolized to produce additional acetyl-CoA and thus have a higher energy yield than glucose [109].

Unlike glucose metabolism, granulosa cells and oocytes basically do not have the function of fatty acid synthesis. Therefore, granulosa cells no longer serve as the donor of oocyte metabolic raw materials in lipid metabolism but regulate oocyte meiosis through the intercellular transmission of signal and energy molecules. The β-oxidation pathway is involved in both granulosa cells and oocytes, and its products can be activated by mutual promotion with key regulators of meiosis. cAMP is a common suppressor of meiosis, and meiotic activation is required to disassemble cAMP. LH and HCG, on the eve of ovulation, activate phosphodiesterase (PDE) in cumulus cells, leading to the breakdown of cAMP and activation of the camp–AMP–PrKA pathway, and this leads to the phosphorylation and loss of function of acetyl-CoA carboxylase (ACAC) located in the mitochondria [57,59,110]. This prevents the metabolism of acetyl-CoA to propylene glycolyl CoA and the inhibition of CPT1, the key enzyme of β-oxidation [110,111]. Thus, this leads to the disinhibition of β-oxidation and the production of large amounts of ATP. The ATP surge provides an energy substrate for meiosis and enhances the promoting effect of cAMP breakdown on meiosis. Due to the similarity of lipid metabolism pathways between granulosa cells and oocytes, the excess cAMP, AMP, and ATP from granulosa cells can enter oocytes through gap junctions to enhance the regulation of oocyte meiosis. For example, the high secretion of cAMP from granulosa cells leads to the arrest of oocytes at metaphase II of meiosis. Although the oocyte exhibits high lipid metabolic activity during meiosis, a study based on the transcriptome of the oocyte during development observed a paradoxical situation, which found that the expression of genes involved in triglyceride metabolism, fatty acid synthesis, and high-density lipoprotein (HDL) remodeling was significantly reduced in the oocyte during the preovulatory period. This may be due to transcriptional arrest caused by structural changes in meiotic chromosomes [109]. Therefore, the lipid metabolism of oocytes at this stage may mainly rely on the regulation of proteomics [112].

In addition to the regulation of meiosis, cumulus cells are important for the regulation of lipid levels inside and outside the oocyte. The long-term accumulation of extracellular fatty acids can lead to toxicity to oocytes, and cumulus cells can secrete stearoyl desaturase to convert fatty acids into nontoxic unsaturated states [113]. For intracellular lipid regulation, the number of lipid droplets decreased, and the development was significantly delayed when the bovine oocytes were cultured alone, without cumulus cells [114]. This may be due to the regulation of cortisol levels by cumulus cells. On the eve of ovulation, the 11-β-hydroxysteroid dehydrogenase in cumulus cells acts synergistically with luteinizing hormone (LH) or human chorionic gonadotropin (hCG) to catalyze the conversion of corticosterone to cortisol [115]. A high concentration of cortisol in the follicle can lead to the upregulation of *HSL* expression and promote the conversion of triglycerides to free fatty acids, accompanied by a large reduction in the number of lipid droplets [110,116]. Large amounts of fatty acids provide sufficient energy for oocyte meiosis.

#### 4.3.2. Steroid Metabolism

Cholesterol is another major lipid feedstock in the follicle and is mainly used in the synthesis of steroid hormones. Because cholesterol is not water-soluble, it depends on the various lipoproteins present in the body for transport. Under normal circumstances, cholesterol is ingested by the body through both exogenous and endogenous pathways and then coated in low-density lipoprotein (VLDL), which comprises triglycerides and apolipoprotein APOB [117,118]. Then, the circulating protein esterase (LDL) remodels VLDL and converts it into denser and smaller low-density lipoprotein (LDL), which is transported through the blood stream to supply each cell [119]. However, the unique multilayered barrier structure of the follicle allows only small molecules to diffuse freely, and LDL is limited by its large volume. Therefore, the cholesterol supply of the follicle is mainly dependent on the smaller volume of HDL. Compared with HDL in normal plasma, this lipoprotein has a relatively lower cholesterol load and more phospholipids and apolipoproteins APOA-1 and APOA-4, thus allowing HDL to undergo structural rearrangement to improve cholesterol absorption efficiency after contact with host cells [120,121]. Before entering granulosa cells, cholesterol undergoes theca cell processing and conversion to androstenedione, a step regulated by luteinizing hormone (LH) [122]. After that, androstenedione crosses the basement membrane and enters the wall granulosa cells (MGCs), where aromatase is activated and mediates the efficient conversion of androstenedione to estrogen under the action of follicle-stimulating hormone (FSH) [122]. In follicular development, estrogen is mainly used to prevent follicular atresia [14]. After ovulation, residual granulosa cells maintain pregnancy by metabolizing cholesterol to progesterone [14].

Although granulosa cell cholesterol metabolic pathways have been extensively studied, the pattern of oocyte cholesterol metabolism and its significance for its own development have not been fully elucidated. At present, it is confirmed that oocytes can indeed achieve cholesterol uptake from granulosa cells and follicular fluid: cholesterol was found to flow into oocytes through granulosa cells at the gap junction between granulosa cells and oocytes [55]. The expression of the lipoprotein receptors *LDLR* and *SCARB1* is also found on the surface of oocytes in metaphase II of meiosis [109]. Meanwhile, a high follicular cholesterol concentration caused by hypercholesterolemia can lead to intra-follicular cholesterol overload. However, in the absence of aromatase, which is highly expressed in oocytes, cholesterol uptake can easily lead to an increase in cholesterol concentration in oocytes. Compared with the unknown positive effect of cholesterol on oocytes, the negative effect of cholesterol concentration surge on oocytes is more serious: the mouse oocyte high-cholesterol model constructed by knocking out SCARB1 cannot reproduce, and this phenomenon can be alleviated by reducing HDL concentration [123]. The possible cause of infertility caused by high cholesterol is an abnormal activation of meiosis [124]. High cholesterol activates maturation-promoting factor (MPF) and mitogen-activated protein kinase (MAPK), leading to the premature activation of meiosis, which cannot be maintained in metaphase II of meiosis before fertilization, thus resulting in the failure of zygote formation.

## 5. Relationship between Abnormal Glucose and Lipid Metabolism of Granulosa Cells and PCOS

### 5.1. Overview of PCOS

Polycystic ovary syndrome (PCOS) is a common endocrine disorder in women which involves the characteristic pathological changes of ovarian tissue. Microscopically, dominant follicles do not appear and are unable to undergo normal ovulation. At the same time, multiple immature follicles undergo abnormal development and accumulate on the ovarian surface; these follicles have a small diameter and present a membrane-wrapped cystic structure [125]. Macroscopically, it manifests as enlarged ovarian volume with prolonged inflammation and fibrosis resulting from hormonal responses. Pallor and irregularity caused by accumulation of abnormal follicles are also present on the surface of the ovary. The direct cause of polycystic ovary is mainly endocrine disorders, usually manifesting as insulin resistance, increased androgen levels, and decreased estrogen and progesterone levels. Abnormal levels of these hormones lead to ovarian dysfunction and interfere with normal follicular development [126]. At the same time, ovarian inflammatory response and genetic factors are also related to the pathogenesis of PCOS [126]. Cumulus cells (CCs) and mural granulosa cells (MGCs) are two major granulosa cells in the follicle which act synergistically and play important functions during oocyte development. The granulosa cells of normal sugar and lipid metabolism have been shown to have great significance for follicular development. Therefore, the abnormal glucose and lipid metabolism of granulosa cells is bound to lead to ovarian development disorders. The pathological state of the glucose and lipid metabolism of granulosa cells and its relationship with PCOS are discussed in the following subsection.

### 5.2. Abnormal Glucose Metabolism of Granulosa Cells in PCOS

In studies of clinical and mammalian models of PCOS, insulin resistance is a key factor mediating abnormal glucose and lipid metabolism in granulosa cells. Insulin resistance refers to the decreased sensitivity of cells to insulin, resulting in decreased glucose uptake and compensatory secretion of insulin, which is manifested as hyperinsulinemia. Insulin resistance and abnormal glucose metabolism of granulosa cells are closely related; together, they lead to PCOS [126]. Insulin resistance by inhibiting insulin activates the P13K-AKT pathways that inhibit granulosa cell lactic acid; lactic acid, as the main granulosa cells to supply oocytes’ generation energy substrate, meet the energy needs of oocyte development, resulting in oocyte energy supply [127]. At the same time, insulin resistance can change the expression of glucose transporter (*GLUT*) on the surface of granulosa cells. An analysis of GLUT on the surface of granulosa cells in normal and pathological conditions revealed differential expressions of *GLUT4*, *6*, *9*, *10*, *11*, and *12*. A logistic analysis confirmed that these differential expressions were correlated with the rates of oocyte fertilization and implantation. The reason for this may be that GLUT changes limit glucose uptake in granulosa cells and lead to oocyte hypoplasia [128]. Excessive androgen can aggravate PCOS in many ways by inducing ovarian cysts and destroying ovarian function. However, whether androgen mediates PCOS by affecting GC glucose metabolism remains controversial. An experiment based on mouse granulosa cells showed that androgen receptor (AR)-binding enzyme phosphoglycerate kinase (PGK1) was highly expressed in PCOS mice, which could regulate glucose metabolism, triggering metabolic disorders in granulosa cells, as well as enhance AR stability and nuclear translocation, thus aggravating cytotoxicity of androgen [129]. However, in another experiment, a high concentration of testosterone was used to treat granulosa cells, and it was found that AKT phosphorylation and lactate production downstream of insulin did not change significantly, indicating that high concentrations of androgen could not directly regulate the glucose metabolism of granulosa cells involved in insulin [127].

Mitochondrial dysfunction is another important cause of abnormal glucose metabolism in granulosa cells. Mitochondria are the material metabolism site of the tricarboxylic acid cycle and oxidative phosphorylation downstream of glucose metabolism; they participate in most of the energy output of glucose metabolism and fatty acid metabolism. By comparing the expression of mitochondrial activity measures between people with PCOS and people without PCOS, researchers found that the level of adenosine triphosphate, mitochondrial membrane potential, and mitochondrial DNA copy number were significantly reduced in granulosa cells and oocytes of people with PCOS, and ROS accumulation occurred [130]. This suggests mitochondrial dysfunction among people with PCOS, as well as oxidative stress in the oocytes. Comprehensive transcriptional profiling of normal and people with PCOS through induced pluripotent stem cell technology (IPSC) revealed a decrease in mitochondrial respiratory capacity and glycolytic function but an increase in mitochondrial copy number in people with IPSC, which may be due to cellular compensation. Metformin, as an IPSC therapeutic agent, can significantly reverse glycolytic function, while rescuing mitochondrial ATP production [131]. In addition, since most of the oxidative phosphorylation process occurs in the oocyte, and granulosa cell activity is partially dependent on the energy production of the oocyte, the downregulation of mitochondrial function often leads to a lack of sufficient energy supply in granulosa cells, leading to the inhibition of glycolysis. In addition, inhibition of mitochondrial activity can cause granulosa cell apoptosis, senescence, and autophagy pathways, which lead to broader granulosa cell dysfunction and further aggravate metabolic disorders.

In the granulosa cells of people with PCOS and animals with PCOS, a variety of endogenous molecular changes have been shown to be associated with insulin resistance and glucose metabolism disorders. Silent information regulator (SIRT3) is significantly downregulated in granulosa cells of people with PCOS. SIRT causes mitochondrial oxidative stress and dysfunction mainly by changing the acetylation status of genes encoding key enzymes of mitochondrial respiratory chain. Downregulation of glucose-6-phosphate dehydrogenase occurs in the SIRT3 knockout mouse model, which triggers the blocking of the glucose metabolism pathway [132]. Growth differentiation factor (GDF8) was found to be expressed in the granulosa cells of people with PCOS and accumulated in follicular fluid. Further studies showed that GDF8 caused glucose metabolism disorders in granulosa cells through the GDF8-ALK5-SMAD-SENRP1 pathway, and this pathway was dependent on TP53. LNK is involved in the regulation of insulin signaling pathway [133]. The expression of *LNK* in granulosa cells of PCOS mice is significantly upregulated, and a high concentration of LNK activates the AKT-FOXO3 signaling pathway and inhibits the transfer of FOXO3 from the nucleus to the cytoplasm, activates insulin resistance and granulosa cell apoptosis, and leads to the downregulation of glucose metabolism [134]. Serum amyloid A1 (SAA1) is an acute-phase protein involved in inflammatory response. In PCOS, SAA1 is feedforward secreted by granulosa cells and reduces GLUT4 membrane translocation and glucose uptake by granulosa cells through PETN and the TLR2/4-NFκB-ATT pathways [135]. Through an analysis of the clinical data of people with PCOS and people without PCOS, an experiment found that there was a high expression of Angiopoietin-like 4 (*ANGPTL4*) in the granulosa cells of people with PCOS, and it was significantly related to glucose and lipid metabolism. However, the specific intrinsic pathway of this remains to be elucidated [136]. Chemerin, an adipokine used to regulate adipogenesis, has recently been found to be highly expressed in people with PCOS and leads to decreased insulin sensitivity and glucose uptake. This is because chemerin interferes with the membrane translocation of GLUT4 by increasing the phosphorylation of substrates and enzymes related to the insulin pathway [137]. The GTPase IMAP family 7 (GIMAP7) is mainly found in granulosa cells and is especially highly expressed in the PCOS rat model. Studies have shown that GIAMP7 can induce oxidative stress by reducing the levels of glutathione (GSH) and superoxide dismutase (SOD) in granulosa cells by inhibiting the sonic hedgehog (SHH) pathway [138]. This is consistent with the induction of oxidative stress via the downregulation of the pentose phosphate pathway due to insulin resistance in PCOS, but it is not known whether GIAMP7 is involved in the regulation of the pentose phosphate pathway.

Altered expression of ncRNA in granulosa cells has also been shown to be associated with PCOS. Vitamin D plays an important regulatory role in ovulation, and a study based on vitamin D-deficient mice showed that miR-196-5p inhibition was significantly downregulated in granulosa cell expression and mediated PCOS. In terms of glucose metabolism, miR-196-5p has been confirmed to inhibit glucose uptake, and the high expression of *GLUT4* on the surface of granulosa cells can be observed by reactivating this mRNA via granulosa cell transfection [139]. IncRNA Inc-CCNL1-3:1(CCNL) is highly expressed in PCOS granulosa cells, and it is mainly through the CCNL-FOXO pathway that we can achieve long-term activation of FOXO in the nucleus, which is related to insulin resistance and apoptosis pathways [140]. In addition, miR-133, miR-223, and miR-93 were significantly upregulated in the cardiomyocytes and subcutaneous adipose tissue of people with PCOS, and they were subsequently confirmed to play an important role in regulating *GLUT4* expression and thereby achieving insulin resistance [141]. However, their expression level in granulosa cells and their role in granulosa cells need to be further explored.

### 5.3. Abnormal Lipid Metabolism of Granulosa Cells in PCOS

Lipid metabolism of granulosa cells in PCOS mainly involves the energy synthesis pathway involving fatty acid metabolism and the steroid hormone production pathway involving cholesterol metabolism. By determining the transcriptomic characteristics of PCOS granulosa cells, researchers have found that granulosa cells are highly expressed in the lipid metabolism pathway, the fatty acid synthesis pathway, and the steroid metabolism pathway [142]. This suggests that granulosa cells have active lipid metabolism. Like glucose metabolism, lipid metabolism is also closely related to insulin resistance. In terms of fatty acid metabolism, hyperinsulinemia induced by insulin resistance promotes the synthesis and storage of fatty acids, while inhibiting the β-oxidation of fatty acids and their uptake and utilization in peripheral tissues. This is confirmed by the upregulated levels of saturated and unsaturated fatty acids in the follicular microenvironment of people with PCOS [143]. For steroid metabolism, glucocorticoid synthesis and insulin resistance show a mutually reinforcing pattern. In insulin-resistant states, a high level of insulin stimulates the expression of *11β-HSD1*, the key enzyme of cortisol synthesis in granulosa cells. High cortisol promotes IR by inhibiting AKT phosphorylation through inducing the expression of the tensin homolog and phosphatase deleted on chromosome 10 [144]. At the same time, cortisol can induce the expression of 1*1β-HSD*1 by regulating IL-1β and can further increase cortisol accumulation [145]. Ovarian myo-inositol (MI) and D-chiral inositol (DCI) have opposite effects: MI is responsible for estrogen production, while DCI mediates androgen synthesis. High concentrations of insulin can promote the conversion of MI into DCI and then produce a large number of androgens [146].

Under pathological conditions, ovarian steroid hormones are characterized by hyposecretion of estrogen and hypersecretion of androgen and progesterone. Granulosa cells play an important role in this pathological state. Granulosa cells are used to down-regulate the aromatase enzyme produced by theca cells that metabolizes androgens to estrogens, leading to a significant androgen accumulation. A variety of external factors have been confirmed to be associated with the low expression of aromatase in granulosa cells. Immature follicles in people with PCOS secrete anti-Mullerian hormone (AMH), which inhibits FSH-induced aromatase activity [147]. An important endogenous hormone, melatonin, is present in the body; long-term exposure to dark environments significantly reduces the granulosa cell surface expression of the melatonin receptor (Mtnr1a), and this decreases the ability of melatonin in regulating glucose metabolism, and also inhibits the expression of the androgen receptor (AR) and CYP19A1, thus reducing the output of aromatase in granulosa cells, hyperandrogenism can occur [148,149]. People with PCOS often experience abnormal adipokine metabolism; this mainly manifests as high levels of leptin and low levels of adiponectin. Among these, leptin can reduce estradiol production by inhibiting aromatase activity [150]. The chemical bisphenol A(BPA) was found to be enriched in the follicular fluid of people with PCOS, and it was confirmed that BPA concentration was negatively correlated with estradiol and aromatase concentrations in a granulosa cell model [151]. Through the detection of testosterone concentration and aromatase mRNA and protein levels in PCOS patients, researchers have found that high testosterone levels in the ovarian environment can inhibit the expression of aromatase; this enriches aromatase and androgen’s mechanism of action [152]. Androgen production is also achieved by granulosa cells themselves: α1AMPK mRNA has low expression in the granulosa cells of people with PCOS. Knocking out α1AMPK in the granulosa cell model can upregulate two key enzymes for androgen synthesis, 3βHSD and P450cscc, and achieve efficient androgen synthesis [153]. A variety of endogenous molecular changes in granulosa cells have been confirmed to affect the secretion of progesterone. Chemerin is highly expressed in the luteinized granulosa cells of people with PCOS; this inhibits progesterone production, which is mediated through chemokine-like receptor 1 (Cmklr1) [154]. C-X-C chemokine ligand 14 (CXCL14) has a low expression in luteinized granulosa cells. The low expression of CXCL14 inhibits JNK and P38 pathways and inhibits the phosphorylation of camp response element binding protein (CREB), thereby downregulating STAR expression and ultimately inhibiting progesterone synthesis [155].

Granulosa cell fatty acid metabolism is mainly responsible for the energy supply of ovarian cells. Hypersecretion of insulin among people with PCOS causes reduced utilization of fatty acids by granulosa cells, leading to cell cycle disorder and dysfunction among granulosa cells and oocytes. Extracellular fatty acid accumulation also causes serious side effects in the follicular microenvironment. High levels of fatty acids can induce the upregulation of inflammatory factors such as IL-6 and IL-8 during transcription and post-translational processing, thereby inducing severe ovarian inflammation [143]. Meanwhile, fatty acid accumulation has been shown to lead to the disorder of steroid metabolic pathways. Monounsaturated fatty acid (MUFA) was found to activate the androgen transcription factors (SOX9) transcription and translation, and the inhibition of estrogen transcription factor FOXL2 [156]. ω-3 polyunsaturated fatty acids activate the P13K-AKT pathway by upregulating CYP51, thereby stimulating estrogen and progesterone biosynthesis [157]. Therefore, the fatty acid metabolism and steroid synthesis pathways of granulosa cells are not independently regulated but may interact with each other to aggravate a pathological state.

Like glucose metabolism, a variety of granulosa cell ncRNAs have been shown to induce dysregulation of granulosa cell lipid metabolism in PCOS. Both miR-93 and miR-21 are androgen dependent and significantly inhibit follicular function under hyperandrogenic conditions. In the regulation of estrogen synthesis in granulosa cells, miR-145 is upregulated in the granulosa cells of people with PCOS, which inhibits estradiol secretion by inhibiting the MAPK/ERK pathway [141]. miR-335-5P is downregulated in PCOS-GCs, leading to the suppression of its ability to increase estradiol production through the SP1-CYP19A1 axis [158]. miR-29a is downregulated in PCOS-GCs, and experiments in granulosa cell models have shown that it regulates GC proliferation and steroidogenesis [159]. miR-320a is downregulated in PCOS-GCs, which leads to the inhibition of the miR-320a/RUNX2/CYP11A1 pathway and the inability to provide adequate steroidogenesis [160]. Under normal conditions, miR-323-3p can inhibit steroidogenesis and GC apoptosis by targeting IGF-1 and its downstream pathways, which are also downregulated in PCOS-GCs [161]. In addition, upregulated miR-27a-3p and downregulated miR-196-5p were found in mouse model experiments [162]. They are important for steroidogenesis, apoptosis, and other aspects of granulosa cells in mice, but their role in human PCOS has not been confirmed. Changes in ncRNAs related to glucose and lipid metabolism in patients with PCOS are shown in Figure 4.

## 6. Discussion

Regulation of metabolic and signaling coupling between oocytes and granulosa cells is important for follicle growth and oocyte maturation. Under normal conditions, the normal glucose and lipid metabolism of granulosa cells produces a large number of energy substrates and signaling molecules, which are efficiently transported through a variety of transport pathways that are established between oocytes and granulosa cells. Abnormal glucose and lipid metabolisms in granulosa cells have been confirmed to be associated with PCOS. The levels of various endogenous molecules and ncRNA related to glucose and lipid metabolism in the granulosa cells of people with PCOS are significantly different from those of people without PCOS. This paper focused on the granulosa cells of sugar metabolism and lipid metabolism and illustrated their significance in the normal development of the follicle; additionally, this study summarized the PCOS-mediated granulosa cell sugar and lipid metabolism disorders of multiple targets. This summary can serve as a support for clinical doctors in diagnosing and treating PCOS, as well as for informing drug development.

Interestingly, although glucose metabolism and lipid metabolism have independent metabolic pathways, they can adapt to environmental changes through intrinsic regulatory mechanisms. By increasing the levels of unsaturated fatty acids (NEFA) in bovine oocytes cultured in vitro, researchers found that oocyte glucose uptake was reduced [163]. A similar phenomenon was observed in mice: glucose consumption decreased following carnitine-stimulated fatty acid oxidation. After the application of the fatty acid oxidation (FAO) inhibitor etomoxir, glucose consumption in the medium was upregulated [164]. This suggests that there is a complementary relationship between oocyte glucose metabolism and lipid metabolism in the energy supply of oocytes. Another study found that, by limiting glucose uptake and providing sufficient levels of fatty acids in oocytes, the proportion of glucose used in energy metabolism was significantly reduced; subsequently, more was found to be used in key pathways, such as those for nucleic acid synthesis and oxidative stress regulation [165]. In the future, we plan to explore the intrinsic pathways of this homeostatic mechanism, study whether the balance between glucose and lipid metabolisms also exists in granulosa cells, and determine whether granulosa cells are involved in the construction of glucose and lipid metabolism homeostasis in oocytes.

## Figures and Tables

**Figure 1 ijms-24-16247-f001:**
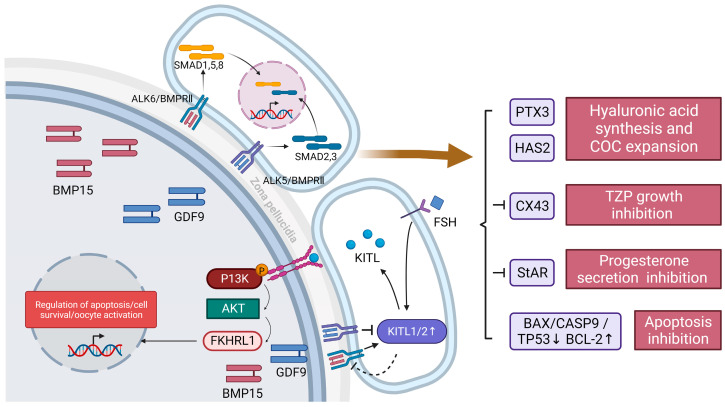
Paracrine-mediated signaling between granulosa cells and oocytes. Oocytes mainly secrete the paracrine factors BMP15 and GDF9. BMP binds to a dimer composed of the TGFβ receptor ALK6/BMPRII and activates SMAD1,5,8. GDF9 binds to a dimer composed of ALK5/BMPR8 and activates SMAD2,3. SMAD controls a variety of biological processes by regulating gene transcription. These include the upregulation of PTX3 and HAS2 expression to promote hyaluronan synthesis and COC expansion; inhibition of CX43 to reduce TZP; inhibition of StAR to reduce progesterone production; regulation of apoptosis-related genes such as BAX, CASP9, TP53, and BCL2 to achieve apoptosis inhibition; and regulation of granulosa cell KITL1 protein levels. KITL is a granulosa cell paracrine factor that activates KIT in response to FSH stimulation and activates the P13K-AKT-FKHRL1 pathway to regulate oocyte survival, activation, and apoptosis. BMP15 promotes the expression of KITL1/2 and is inhibited by high concentrations of KITL. GDF9 inhibits KITL expression. BMP15—bone morphogenetic protein 15; GDF9—growth differentiation factor 9; SMAD—Sma- and Mad-related proteins; PTX3—pentraxin 3; HAS2—hyaluronan synthase 2; CX43—connexin 43; StAR—steroidogenic acute regulatory protein; KITL—stem cell factor (also known as Kit ligand); FSH—follicle-stimulating hormone; FKHRL1—Forkhead box protein O1 (FOXO1).

**Figure 2 ijms-24-16247-f002:**
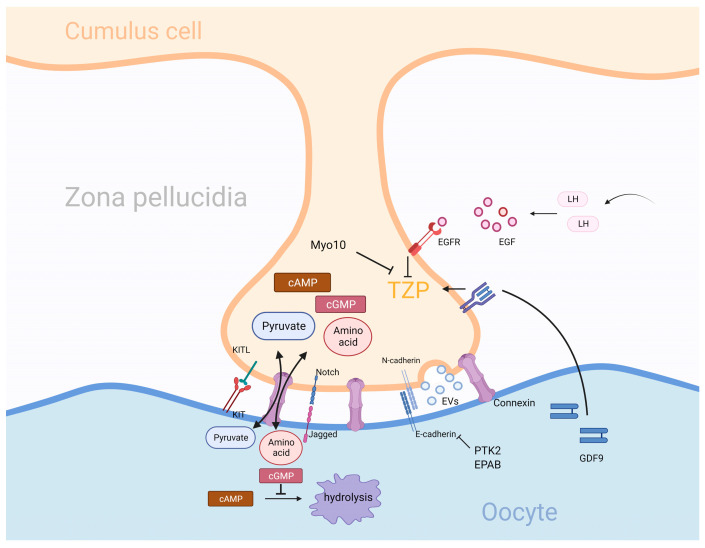
There are three main transport modes for substances at the tip of TZPs: gap junctions, adhesion junctions, and extracellular vesicle transport. Gap junctions enable the efficient transport of substances such as pyruvate, amino acids, cAMP, and cGMP via Connexin. Adhesion junctions mainly include several kinds of E-cadherin/N-cadherin, Notch/Jagged, and KITL/KIT, which are mainly used for stabilizing TZPs structure and intercellular information exchange. Extracellular vesicles are mainly used for macromolecule transport. TZPs are simultaneously regulated by the endogenous granulosa cell signal Myo10; the oocyte signal GDF9; and the extracellular signals EGF, PTK2, and EPAB. TZPs—transzonal projections; GDF9—growth differentiation factor 9; EGF—epidermal growth factor; Myo10—myosin X; LH—luteinizing hormone; EVs—extracellular vesicles; PTK2—proline-rich tyrosine kinase 2; EPAB—eukaryotic poly(A)-binding protein; cAMP—cyclic adenosine monophosphate; cGMP—cyclic guanosine monophosphate.

**Figure 3 ijms-24-16247-f003:**
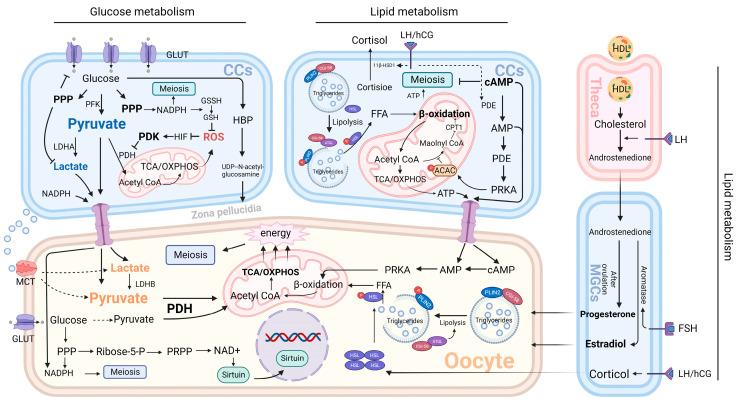
Effects of granulosa cell glucose and lipid metabolism on oocytes. Glucose metabolism in granulosa cells mainly includes glycolysis, the pentose phosphate pathway, and the hexosamine biosynthesis pathway. Granulosa cells deliver energy precursors such as pyruvate and lactate to granulosa cells through glycolysis. GSH and NADPH are supplied to oocytes through the pentose phosphate pathway, which contributes to oxidative stress inhibition and mitotic regulation. Hyaluronic acid dissection is stabilized through the hexosamine biosynthesis pathway to maintain oocyte energy supply. Granulosa cell lipid metabolism is mainly divided into fatty acid metabolism and steroid metabolism. For fatty acid metabolism, granulosa cell β-oxidation products and oocyte meiosis-related factors promote each other to meet the meiotic energy supply. For steroid metabolism, granulosa cells can metabolize estrogen, progesterone, cortisol, and other hormones to achieve oocyte growth regulation. MCG—mural granulosa cell; PFK—phosphofructokinase; PDK—pyruvate dehydrogenase kinase; PDH—pyruvate dehydrogenase; HIF—hypoxia-inducible factor; GSH—glutathione; GSSH—glutathione disulfide; ROS—reactive oxygen species; NADPH—nicotinamide adenine dinucleotide phosphate; TCA—tricarboxylic acid cycle; OXPHOS—oxidative phosphorylation; HBP—hexosamine biosynthetic pathway; LDHA—lactate dehydrogenase A; LDHB—lactate dehydrogenase B; PRPP—phosphoribosyl pyrophosphate; Sirtuin—silent information regulator 2 (Sir2) proteins; GLUT—glucose transporter; MCT—monocarboxylate transporter; PRKA—protein kinase A; ACAC—acetyl-CoA carboxylase; CPT1—carnitine palmitoyltransferase 1; FFA—free fatty acid; PDE—phosphodiesterase; HSL—hormone-sensitive lipase; CGI-58—comparative gene identification 58; ATGL—adipose triglyceride lipase; PLIN2—perilipin 2; HDL—high-density lipoprotein.

**Figure 4 ijms-24-16247-f004:**
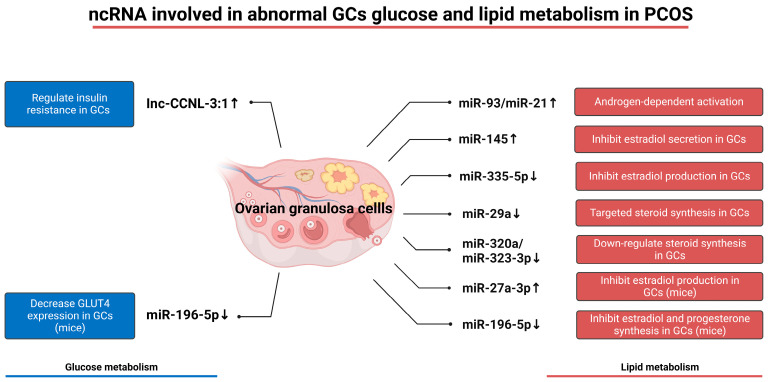
The ncRNA involved in abnormal GCs glucose and lipid metabolism in PCOS. PCOS patients show altered levels of several ncRNAs. miR-196-5p and Inc-CCNL-3:1 affected glucose metabolism in GCs, mainly including glucose uptake and insulin resistance. miR-93, miR-21, miR-145, miR-335-5P, miR-29a, miR-320a, miR-323-3p, miR-27a-3p, and miR-196-5p affect lipid metabolism in GCs, mainly including the synthesis and secretion of multiple steroid hormones. The effects of miR-27a-3p and miR-196-5p on glucose and lipid metabolism have only been identified in mice models and have not been clinically confirmed. GCs—granulosa cells.

## Data Availability

Not applicable.

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
