# Peer review of "Essential Role of Granulosa Cell Glucose and Lipid Metabolism on Oocytes and the Potential Metabolic Imbalance in Polycystic Ovary Syndrome"

_ijms, 2023, doi:10.3390/ijms242216247_

Round 1
Reviewer 1 Report
Comments and Suggestions for Authors
The manuscript titled "Essential role of granulosa cell glucose and lipid metabolism on oocytes and the potential metabolic imbalance in PCOS," authored by Zhang and colleagues, which has been submitted for review, represents a significant contribution to the current understanding of this subject, particularly in the context of PCOS, given the importance of granulosa cells in oocyte maturation quality.
However, as a reviewer, I must point out certain shortcomings that need to be addressed and/or supplemented.
In Chapter 2, subsection 2.1, "Follicular growth," lines 61-66: it is crucial to correct the inaccurate information that suggests the zona pellucida exists between the follicles and granulosa cells. The zona pellucida actually forms between the primary oocytes and granulosa cells at this stage of follicular development.
Lines 67-81: I understand the necessity of presenting the stages of folliculogenesis in a condensed manner; however, several pieces of information here could potentially mislead the reader. Please also incorporate appropriate scientific terminology when describing ovarian follicles (such as basement membrane, theca interna, theca externa, etc.).
Lines 94-96: In the description of the developing corpus luteum, it is essential to consider not only large luteal cells (derived from granulosa cells) but also to include small luteal cells that originate from the theca interna cells.
In subsection 2.2, if the authors mention the mural granulosa layer, it is important to also include the antral layer adjacent to the antrum (lines 98-103). At this point, it is crucial to remember (and incorporate into the manuscript based on literature data) that a prerequisite for ovulation is the pre-ovulatory switch in steroid production. In the pre-ovulatory follicle, granulosa cells begin to produce increasing amounts of progesterone, which not only influences biochemical changes in the cumulus-oocyte complex (COC) but also leads to morphological changes in the follicle, such as the formation of the stigma.
Line 104: The phrase "Through cell deformation..." is exceptionally awkward. Please use more scientific terminology and phrasing.
The figures are praiseworthy; however, both in Figure 2 and Figure 2, the zona pellucida is missing, especially considering that the authors have marked the TZP. Additionally, the title of section 4, "Glucose and lipid metabolism in granulosa cells and oocytes in normal body," should be revised. What does "normal body" mean in this context?
In summary, the paper can be accepted for publication after major revision. Language editing is also necessary.
Comments on the Quality of English Language
Language editing is necessary because the authors frequently use colloquial and non-scientific language.
Reviewer 2 Report
Comments and Suggestions for Authors
The authors have submitted a very thorough comprehensive article for review. The article sufficiently provided introduction to the subject, and discussed the communication and material exchange between oocytes and granulocytes well with emphasis on PCOS. The communication and material exchange between the granulocytes, cumulus cells, theca cells and oocytes are complex and tightly regulated to ensure the proper balance and functioning of the system. The article adequately summarizes the current literature on the metabolism and communication between GCs and oocytes. The only concern is the length of the article, which is too elaborate. The authors could consider making it concise for the readers to follow effectively.
Reviewer 3 Report
Comments and Suggestions for Authors
A huge work was performed by the Authors of this paper.
The manuscript is presented in communicative and easy to read manner. However due to the subject there is plenty of associations which might be lost during reading because of missing figures. There are 3 figures but in my opinion it is significantly to less as well those which were used are very complex and even they are proper and precise they are difficult to memorize overview. I would recommend introduction few simplified graphs to make it more convenient for remembering.
There are minor editorial mistakes which may be easily fixed during final proof reading.
Moreover I would think about shortening of the references, as there are some papers which are really old ones, but I leave this decision up to the Authors will.
